# Learn to Learn Consistently via Meta Self-distillation for Few-shot Classification

## Abstract

In few-shot learning, a model trained on disjoint base classes must solve novel tasks at test time using only a few of examples. A central challenge is shortcut bias: the model can overfit to spurious cues (e.g., background, noise, shape, color) that separate the few support examples during rapid adaptation but fail to generalize to larger query sets within novel tasks. In this paper, we first define learning consistency (i.e., the degree to which the model acquires similar knowledge when trained from different views of the same data), and show empirically that higher consistency reduces reliance on shortcuts and improves generalization. Building on this insight, we propose Learn to Learn Consistently (LLC), a simple yet effective meta-learning method that maximizes learning consistency during training. In the inner loop, the model is updated separately using different augmented views of the same support set. In the outer loop, the same query set is used to enforce consistency across the learned updates. Models initialized by LLC generalize better in the meta-testing phase. Extensive experiments demonstrate improved generalization across diverse settings and stronger learning consistency.

## 1 Introduction

Few-shot learning aims to address novel tasks with a limited number of examples, typically through fast adaptation of a model trained on a dataset with disjoint labels. Many approaches tackle this issue from the perspective of meta-learning (Lee et al., 2019; Lake & Baroni, 2023; Wei et al., 2024c). Methods such as Model-Agnostic meta-Learning (MAML) (Finn et al., 2017) and its variants (Nichol et al., 2018; Raghu et al., 2019; Kao et al., 2021; Wei et al., 2024b) aim to learn initialized parameters for a model with prior knowledge for fast adaptation. Recent research has explored more challenging scenarios, such as cross-domain few-shot learning (Tseng et al., 2020; Guo et al., 2020; Ullah et al., 2022; Wei et al., 2024a), where the novel task belongs to a different domain and label set than the training dataset.

A key challenge in various few-shot learning problems is the model's tendency to learn biased shortcut features (e.g., background, noise, shape, color) from limited examples (Shah et al., 2020; Lyu et al., 2021; Le et al., 2021; Teney et al., 2022). These shortcut features may suffice to distinguish the few classes during fast adaptation but result in poor generalization. Several solutions have been proposed to address these issues. Although these approaches partially mitigate the problem, they often require additional resources or learn generalized features only within the meta-train dataset (Snell et al., 2017; Liu et al., 2020; Le et al., 2021; Zhou et al., 2023). From the perspective of meta-learning, we ask the following question: *Can we bias the initialization toward learning generalizable rather than shortcut features on novel tasks?*

This problem is challenging to address directly, as identifying generalized versus shortcut features in the data is difficult. In our study, we generate different views of the same data through data augmentation, which gives these views different shortcut features but similar generalized features. We use these views to update the model and observe that when model learning with better consistency tends to exhibit better generalization. This implies that when a model learned with higher consistency, it would be less influenced by the shortcut features and reaches higher accuracy. At this point, if we can enhance the model's learning consistency across all tasks, we can make the model less influenced by the shortcut feature and more inclined to learn generalized features.

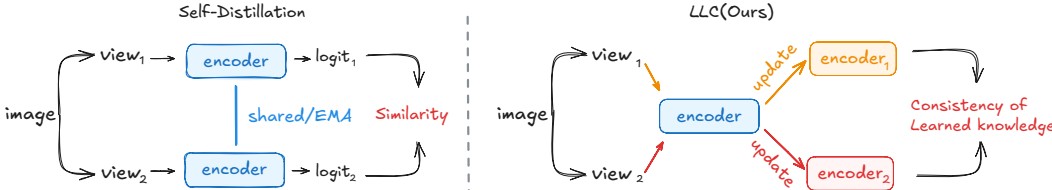

Figure 1: **The core idea between self-distillation and meta self-distillation.** Self-distillation aims to make the deep representation of different views closer, while LLC aims to learn consistent knowledge from the different views of the same image.

Based on this observation, we introduce an optimization-based meta-learning framework, termed Learn to Learn Consistently (LLC), which is designed to maximize the learning consistency of the meta-initialized model. The core idea is to encourage the model to produce stable and coherent predictions across different learning trajectories derived from the same tasks. Concretely, in the inner loop, we apply data augmentation to generate multiple variants of the same tasks, and use these augmented tasks to independently update the initialized model parameters. This process simulates diverse learning paths while preserving the underlying task semantics. In the outer loop, we explicitly optimize for output consistency by minimizing the discrepancy between the predictions on identical query samples, obtained from the differently updated models in the inner loop. By aligning the outputs across these independent updates, LLC strengthens the model's capability to learn in a stable and reliable manner, even when faced with variations in task presentation. This consistency-driven optimization not only enhances the robustness of the meta-initialized model but also improves its generalization ability to unseen tasks. We conduct extensive evaluations of LLC under three distinct few-shot learning scenarios, covering both classification and regression settings. The experimental results consistently demonstrate that LLC outperforms baseline meta-learning methods, validating its effectiveness in promoting stable learning dynamics and superior generalization performance.

In summary, our contributions are as follows:

- We first propose the concept of *learning consistency* and observed that the learning consistency could serve as an indicator of the model's inclination towards learning shortcut features that lead to overfitting.

- Moreover, we propose LLC, which maximizes the consistency of learned knowledge during meta-training, biasing the initialization toward generalizable features.

- Extensive experiments demonstrate that our method achieves remarkable performance across various few-shot scenarios and significantly enhances the model's ability to learn consistently in unseen tasks.

## 2 RELATED WORK

### 2.1 FEW-SHOT LEARNING AND META-LEARNING

Few-shot learning is a compelling paradigm for recognizing novel classes from a limited number of examples. In the conventional setting, a model is first pre-trained on a set of base classes and then rapidly adapted to novel tasks, whose classes are disjoint from those in the pre-training phase. Two main strategies have been pursued in recent studies: transfer learning and meta-learning. Transfer learning methods focus on developing a robust feature extractor from the base classes that can generalize to novel tasks (Tian et al., 2020; Liu et al., 2021; Wei & Wei, 2024), whereas meta-learning—often referred to as learning to learn—aims to endow the model with prior knowledge that enables fast adaptation (Antoniou et al., 2018; Ye & Chao, 2021; Hu et al., 2025).

Meta-learning methods can be generally categorized into metric-based and optimization-based approaches. Metric-based methods, such as ProtoNet (Snell et al., 2017), enhance the feature space by minimizing the distance between support and query examples of the same class, thereby reducing or even obviating the need for fine-tuning at test time. Optimization-based approaches, including MAML (Finn et al., 2017) and its extensions such as Unicorn-MAML (Ye & Chao, 2021), MAML++(Antoniou et al., 2018), and ANIL(Raghu et al., 2019), aim to learn initialization param-

eters that enable efficient adaptation to novel tasks. However, these variants primarily focus on improving the optimization stability of MAML rather than further enhancing the learning ability of the initialized model.

A common challenge in few-shot learning is the tendency of models to latch onto shortcut features during fast adaptation, which leads to overfitting and reduced generalization performance (Lyu et al., 2021; Le et al., 2021; Teney et al., 2022; Wei et al., 2025a). This issue is exacerbated in cross-domain few-shot learning scenarios, where base and novel classes originate from different domains (Triantafillou et al., 2019; Tseng et al., 2020; Ullah et al., 2022). Several remedies have been proposed to solve the problem. For example, Poodle (Le et al., 2021) leverages additional data to penalize out-of-distribution samples, while LDP-net (Zhou et al., 2023) employs both local and global knowledge distillation to encourage the learning of diverse features. However, these approaches typically require extra data or increased model complexity. In this work, we solve the issue by explicitly biasing the initialized model toward learning generalized features.

## 2.2 SELF-DISTILLATION

Self-distillation (Zhang et al., 2021; Wei et al., 2025b) has recently gained attention as a variant of contrastive learning, aiming to refine feature representations by aligning the outputs of positive instance pairs without relying on explicit negative pairs. Methods such as BYOL (Grill et al., 2020) utilize an exponential moving average of the network to generate targets for an online network, while SimSiam (Chen & He, 2021) investigates the mechanisms that prevent representational collapse in the absence of negative examples. Additionally, work by (Allen-Zhu & Li, 2020) suggests that self-distillation can act as an implicit ensemble mechanism, enhancing the discrimination of subtle feature variations. Different from self-distillation that directly aligns representations, our approach focuses on aligning the models differently updated by different augmented Samples. In other words, we focus on learning consistency instead of representation consistency, which aims to reduce the influence of shortcut features during adaptation to novel tasks.

## 3 PRELIMINARY

**Problem Definition For Few-shot Classification**  We define the few-shot classification problem (FSL) as an $\mathcal{N}$-way $\mathcal{K}$-shot task, where there are $\mathcal{N}$ classes, each containing $\mathcal{K}$-labeled support samples. Typically, $\mathcal{K}$ is small, such as 1 or 5. The data used to attempt to update the model is defined as the support data $\mathcal{S} = \{x_s, y_s\}$, where each $x_s$ represents the model's input, and $y_s$ denotes the corresponding label for $x_s$. The data used to evaluate the effectiveness of the model updates is defined as the query data $\mathcal{Q} = \{x_{\mathcal{Q}}, y_{\mathcal{Q}}\}$, which has the same class as the support data, but the samples contained in the query set are different from those in the support set. The FSL task is defined as the problem of learning to correctly classify the query data $\mathcal{Q}$ with the support data $\mathcal{S}$, which can be written as follows:

$$\arg\min_{\theta} \mathbb{E}_{\mathcal{S},\mathcal{Q}\sim\mathcal{D}_{\text{meta-test}}} \left[\mathcal{L}_{\text{FSL}}(\theta, \mathcal{S}, \mathcal{Q})\right] \tag{1}$$

If the model is randomly initialized and directly fine-tuned on the limited support data, the model will overfit. To address that, we need to transfer knowledge from seen data to the unseen data. The seen data used in FSL is referred to as the meta-train set, and the unseen data is referred to as the meta-test set. The labels in the two sets are disjoint, and in the cross-domain few-shot learning, the domains of the two sets are also different.

**Model-Agnostic Meta-Learning**  The objective of MAML is to learn initialized parameters $\theta$ with prior knowledge, such that after a few steps of standard training on the support data, the model can generalize well on the query data:

$$\arg\min_{\theta} \mathbb{E}_{\mathcal{S},\mathcal{Q}\sim\mathcal{D}_{\text{meta-train}}} \left[\mathcal{L}(\mathcal{U}^k(\theta, \mathcal{S}), \mathcal{Q})\right], \tag{2}$$

where $\mathcal{U}^k$ denotes $k$ updates of the parameter $\theta$ using tasks sampled from the task distribution, which corresponds to adding a sequence of gradient vectors to the initialized parameters:

$$\mathcal{U}^k(\theta, \mathcal{S}) = \theta - \sum_{i=1}^{k} \alpha \cdot \frac{\partial\mathcal{L}(\mathcal{U}^{i-1}(\theta, \mathcal{S}), \mathcal{S})}{\partial\theta}, \tag{3}$$

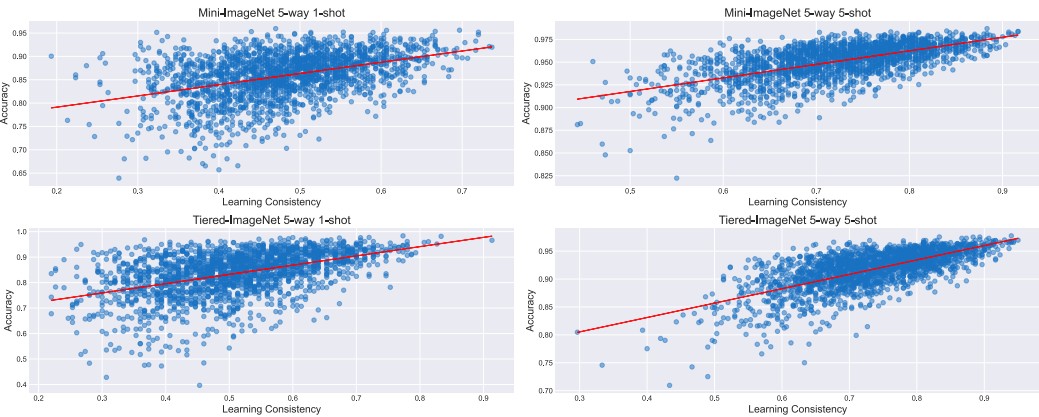

Figure 2: **The learning consistency versus accuracy of the model initialized by MAML across different tasks.** The results demonstrate a clear trend: as the model's consistency learned from the task increases, the average accuracy in predicting query data improves. The algorithm flow to get the result is shown in Supplementary Material Algorithm 1

where $\mathcal{U}^0(\theta, \mathcal{S}) = \theta$. The process of updating the parameters with support data is referred to as *inner loop process*, where $\alpha$ is the stepsize of the inner loop. Subsequently, the query data $\mathcal{Q}$ is used to evaluate $\mathcal{U}^k(\theta, \mathcal{S})$, and directly updating the initial parameters $\theta$, which known as the *outer loop process*. The outer loop commonly employs SGD for updates, and the update process can be computed as follows:

$$\theta' = \theta - \beta \cdot \frac{\partial \mathcal{L}(\mathcal{U}^k(\theta, \mathcal{Q}), \mathcal{S})}{\partial \theta}, \tag{4}$$

where $\beta$ is the learning rate of the outer loop. By minimizing the loss across sampled tasks, MAML enables the parameters to learn prior knowledge from the meta-train set.

## 4    LEARN TO LEARN CONSISTENTLY

### 4.1    WHY LEARN CONSISTENTLY IN FSL

Previous studies have indicated that in few-shot learning (FSL) scenarios, models tend to learn shortcut features (e.g., background, noise, shape, color) from limited examples (Shah et al., 2020; Teney et al., 2022; Lyu et al., 2021; Le et al., 2021). These shortcut features may suffice to distinguish the few classes during rapid adaptation but often lead to poor generalization. From the perspective of meta-learning, we aim for the initialized model to learn more generalized features, avoiding the reliance on shortcut features. However, it's hard to distinguish these features directly in practice. To solve that, We introduced the concept of learning consistency and proposed that the learning consistency can serve as an indicator of the model's inclination towards learning shortcut features.

**Definition 1** (Learning Consistency)**.** *Let $\mathcal{S}$ be a given support sample set and $\theta$ be an initial model. Consider two random data augmentation operators $\mathcal{A}ug_i$ and $\mathcal{A}ug_j$ that generate two augmented versions of the source data: $\mathcal{A}ug_i(\mathcal{S}), \mathcal{A}ug_($\mathcal{S})$ These augmented data are then used to update the initial model $\theta$ to yield two updated models $\theta^i$ and $\theta^j$:*

$$\theta^i \leftarrow \mathcal{U}^k(\theta, \mathcal{A}ug^i(\mathcal{S})), \quad \theta^j \leftarrow \mathcal{U}^k(\theta, \mathcal{A}ug^j(\mathcal{S})).$$

*Given a query sample set $x_\mathcal{Q} \sim p(x_\mathcal{Q})$, let $f_{\theta^i}(x)$ and $f_{\theta^j}(x)$ denote the outputs of the two models, respectively. Define a discrepancy metric $\mathcal{F}_{sim}$ to measure the difference between these outputs. The learning consistency of initialized model $\theta$ is defined as:*

$$\mathrm{LC}(\theta) = \mathbb{E}_{x_\mathcal{Q} \sim p(x_\mathcal{Q}), \mathcal{S} \sim p(\mathcal{S})}[\mathcal{F}_{sim}(f_{\theta^i}(x_\mathcal{Q}), f_{\theta^j}(x_\mathcal{Q}))].$$

Since the support data are differently augmented from the same data, they would contain different shortcut features and similar generalized features. Therefore, the inconsistency of the differently updated models is mainly caused by different shortcut features. If the model tends to learn shortcut features, which results in overfitting, the inconsistency in these features leads to greater output

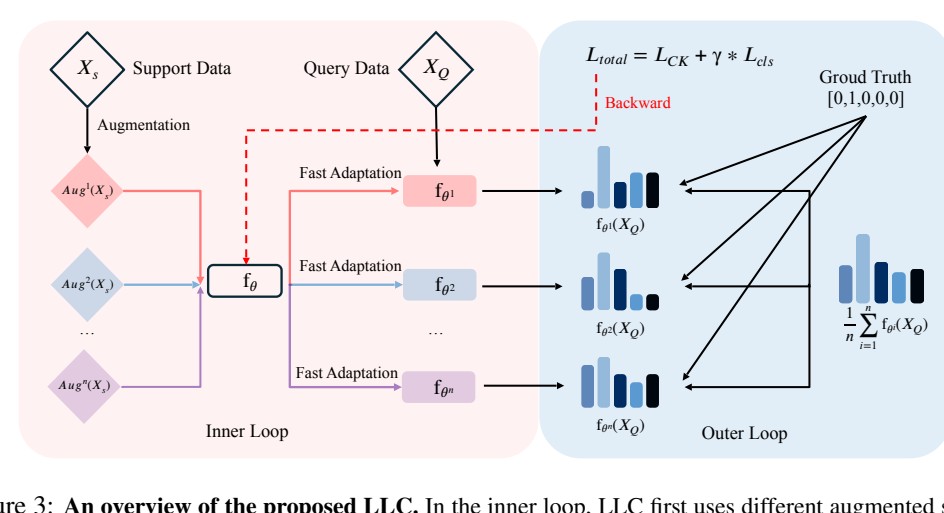

Figure 3: **An overview of the proposed LLC.** In the inner loop, LLC first uses different augmented support data to update the $f_\theta$. In the outer loop, then maximizes the consistency among the outputs of the same query data with different update versions of the initial model

variance for the same query data. To further validate the point, we evaluate the learning consistency and accuracy of the parameters initialized by MAML. The results are illustrated in Figure 2: lower consistency corresponds to lower average prediction accuracy. Additionally, as the amount of support data increases, the model is less influenced by shortcut features, and both consistency and accuracy are improved consistently. These empirical findings indicate a strong alignment between learning consistency and model generalization, thereby substantiating the claim that learning consistency reflects the model's inclination toward shortcut feature learning. We then define the learning consistency property as follows:

**Property 1**(Learning consistency property.) *Given a support sample set $\mathcal{S}$, inner step $k$, input domain $\mathcal{D}$, random augmentation $\mathcal{A}ug^i$ and similarity function $\mathcal{F}_{sim}$. We say initialized parameters $\theta$ satisfy the learning consistency property if:*

$$\forall i, j, \forall \mathcal{S} \in \mathcal{D}_\mathcal{S}, x_\mathcal{Q} \in \mathcal{D}_\mathcal{Q},$$
$$\mathcal{F}_{\text{sim}}(f_{\theta^i}(x_\mathcal{Q}), f_{\theta^j}(x_\mathcal{Q})) = \mathcal{M} \tag{5}$$

where $\theta^i = \mathcal{U}^k(\theta, \mathcal{A}ug^i(\mathcal{S}))$ and $\mathcal{M}$ is the upper bound of $\mathcal{F}_{\text{sim}}$. In short, if the initialized parameters satisfy the learning consistency property, the parameters will learn consistent knowledge from different augmented views of the same image, thus less influenced by the shortcut features.

## 4.2 LEARN TO LEARN CONSISTENTLY

Motivated by the property in Eq. 5, we propose Learn to Learn Consistently (LLC) to enhance the learning consistency of the model initialization. An overview of LLC is provided in Fig. 3.

**Meta-Train Phase.** Specifically, we sample tasks from the meta-train set to obtain support and query data. Unlike MAML, which samples multiple tasks, we sample a single task and create multiple augmented versions as substitutes. Only the support data is augmented in the different augmented tasks, and the tasks share the same query data. The rationale behind this is to have the same standard when assessing the knowledge learned by the model. In the inner loop, the model is updated with different augmented views of the support data to obtain varied parameters: $\theta_i = \mathcal{U}^k(\theta, \mathcal{A}ug^i(\mathcal{S}))$. In the outer loop, we test the query with different updated parameters. The consistency of the query outputs is used to assess the consistency of learned knowledge, considering there are more than two tasks, we use the average of the output to align the learning consistency of the initialized model. The learning consistency loss can be formulated as follows:

$$\mathcal{L}_{\text{CK}} = -\frac{1}{n} \sum_i^n \mathcal{F}_{\text{sim}}(f_{\theta^i}(x_\mathcal{Q}), \frac{1}{n} \sum_j^n f_{\theta^j}(x_\mathcal{Q})) \tag{6}$$

We use the cosine similarity as the similarity function in practice. Furthermore, to ensure the model fully utilizes label information, we also compute the classification loss for each updated parameter

by query data. The model's total loss is expressed as:

$$\mathcal{L}_{\text{total}} = \mathcal{L}_{\text{cls}} + \gamma \cdot \mathcal{L}_{\text{CK}} \tag{7}$$

Where $\gamma$ represents the coefficient of consistency loss. The process of updating the initial parameters is $\theta' = \theta - \beta \cdot \nabla_\theta \mathcal{L}_{\text{total}}$. Where $\beta$ represents the learning rate in the outer loop. The detailed algorithm flow is shown in Supplementary Material Algorithm 2.

**Meta-Test Phase.** During the meta-test phase, LLC is consistent with MAML, which performs fast adaptation support data using SGD and classifies the query data directly with the updated model.

## 5 EXPERIMENT

### 5.1 EXPERIMENT SETTING

**Datasets.** For standard and augmented FSL evaluation, Our method was primarily evaluated on two benchmark datasets: Mini-ImageNet (Vinyals et al., 2016) and Tiered-ImageNet (Ren et al., 2018), both widely used for few-shot learning assessments. For cross-domain FSL evaluation, we use Mini-ImageNet as the source domain and use another eight datasets as the target domain, i.e., CUB, Cars, Places, Plantae, ChestX, ISIC, EuroSAT and CropDisease.

The Mini-ImageNet dataset comprises 100 classes, each containing 600 samples. Following prior work, we divided the 100 classes into training, validation, and test sets, containing 64, 16, and 20 classes, respectively. The Tiered-ImageNet dataset encompasses 608 fine-grained classes, which are categorized into 34 higher-level classes. In alignment with previous studies, we divided these higher-level classes into training, validation, and test sets, comprising 20, 6, and 8 higher-level classes, respectively. CUB, Cars, Places, and Plantae proposed in (Tseng et al., 2020) contain natural images of different properties. ChestX, ISIC, EuroSAT and CropDisease proposed in (Guo et al., 2020) are cross-domain datasets from the domain of medicine, agriculture, and remote sensing, which have significant domain shifts. All the images are resized to $84 \times 84$ pixels following common practice.

**Backbone Model.** For our model evaluation, following (Lee et al., 2019), we employed a ResNet-12 (He et al., 2016) architecture, noted for its broader widths and Dropblock modules as introduced by (Ghiasi et al., 2018). This backbone is broadly used across numerous few-shot learning algorithms. Additionally, we follow the original MAML approach, utilizing a 4-layer convolutional neural network(Conv4) (Vinyals et al., 2016). Following the recent practice (Ye et al., 2020; Qiao et al., 2018; Rusu et al., 2018), MAML, Unicorn-MAML, and our models' weights are pre-trained on the meta-train set to initialize.

### 5.2 MAIN RESULTS

We evaluate our method under three settings: standard few-shot learning problems, cross-domain few-shot learning problems, and augmented few-shot learning problems.

#### 5.2.1 STANDARD FEW-SHOT LEARNING PROBLEMS.

The results in Table.1 demonstrate the performance of LLC and several mainstream few-shot algorithms on few-shot tasks. LLC exhibits a significant improvement over MAML in standard few-shot tasks. The results of MAML are produced by(Ye & Chao, 2021), which uses more inner steps for MAML to reach better performance. On Mini-ImageNet, our method achieved an increase of 0.99% in 5way-1shot and 1.44% in 5way-5shot tasks compared with MAML, respectively. On Tiered-ImageNet, the improvements for 5way-1shot and 5way-5shot tasks were 2.79% and 2.50% compared with MAML, respectively. LLC shows excellent effectiveness in few-shot tasks, with better performance compared to the recent meta-learning algorithms and MAML's variants.

#### 5.2.2 CROSS DOMAIN FEW-SHOT LEARNING PROBLEMS.

To explore the performance when there is a large domain gap between the meta-train set and the meta-test set, we also evaluated the performance of LLC under the cross-domain dataset setting. The results are shown in Table.2. Experimental results demonstrate that our method achieves significant

Table 1: **5way-1shot and 5way-5shot classification accuracy in standard few-shot classification task** and 95% confidence interval on Mini-ImageNet, Tiered-ImageNet (over 2000 tasks), using ResNet-12 as the backbone. NIW-Meta used ResNet-18 as the backbone.

| | Mini-ImageNet | | Tiered-ImageNet | |
|---|---|---|---|---|
| Methods | 1-Shot | 5-Shot | 1-Shot | 5-Shot |
| ProtoNet (Snell et al., 2017) | $62.39 \pm 0.20$ | $80.53 \pm 0.20$ | $68.23 \pm 0.23$ | $84.03 \pm 0.16$ |
| MAML (Finn et al., 2017) | $64.42 \pm 0.20$ | $83.44 \pm 0.14$ | $65.72 \pm 0.20$ | $84.37 \pm 0.16$ |
| MetaOptNet (Lee et al., 2019) | $62.64 \pm 0.35$ | $78.63 \pm 0.68$ | $65.99 \pm 0.72$ | $81.56 \pm 0.53$ |
| ProtoMAML (Triantafillou et al., 2019) | $64.12 \pm 0.20$ | $81.24 \pm 0.20$ | $68.46 \pm 0.23$ | $84.67 \pm 0.16$ |
| DSN-MR (Simon et al., 2020) | $64.60 \pm 0.72$ | $79.51 \pm 0.50$ | $67.39 \pm 0.82$ | $82.85 \pm 0.56$ |
| Meta-AdaM (Sun & Gao, 2024) | $59.89 \pm 0.49$ | $77.92 \pm 0.43$ | $65.31 \pm 0.48$ | $85.24 \pm 0.35$ |
| LA-PID (Yu et al., 2024) | $63.29 \pm 0.48$ | $79.18 \pm 0.43$ | $64.77 \pm 0.47$ | $82.59 \pm 0.37$ |
| NIW-Meta[†] (Kim & Hospedales, 2024) | $65.49 \pm 0.56$ | $81.71 \pm 0.17$ | $70.52 \pm 0.19$ | $85.83 \pm 0.17$ |
| LLC | $\mathbf{65.41 \pm 0.47}$ | $\mathbf{84.88 \pm 0.29}$ | $\mathbf{68.51 \pm 0.53}$ | $\mathbf{86.87 \pm 0.34}$ |

Table 2: **5way-5shot classification accuracy in cross-domain few-shot classification task** (over 2000 tasks), using ResNet-12 as the backbone. Only the train set of Mini-ImageNet is used during the meta-train phase.

| | CUB | Cars | Places | Plantae | Euro | ISIC | CropD | ChestX |
|---|---|---|---|---|---|---|---|---|
| GNN (Garcia & Bruna, 2017) | 62.87 | 43.70 | 70.91 | 48.51 | 78.69 | 42.54 | 83.12 | 23.87 |
| GNN+FT (Tseng et al., 2020) | 64.97 | 46.19 | 70.70 | 49.66 | 78.02 | 40.87 | 87.07 | 24.28 |
| TPN+ATA (Wang & Deng, 2021) | 70.14 | 55.23 | 73.87 | 59.02 | 85.47 | 49.83 | 93.56 | 24.74 |
| GNN+ATA (Wang & Deng, 2021) | 66.22 | 49.14 | 75.48 | 52.69 | 83.75 | 44.91 | 90.59 | 24.32 |
| MatchingNet+AFA (Hu & Ma, 2022) | 59.46 | 46.13 | 68.87 | 52.43 | 69.63 | 39.88 | 80.07 | 23.18 |
| GNN+AFA (Hu & Ma, 2022) | 68.25 | 49.28 | **76.21** | 54.26 | 85.58 | 46.01 | 88.06 | 25.02 |
| LDP-net (Zhou et al., 2023) | **70.39** | 52.84 | 72.90 | 58.49 | 82.01 | 48.06 | 89.40 | 26.67 |
| GNN +FAP (Zhang et al., 2024) | 67.66 | 50.20 | 74.98 | 54.54 | 82.52 | 47.60 | 91.79 | 25.31 |
| RFS+MLP (Bai et al., 2024) | - | - | - | - | 83.14 | 46.02 | 66.87 | **29.09** |
| LLC | 70.22 | **58.55** | 75.59 | **60.81** | **85.65** | **51.54** | **95.12** | 28.26 |

Table 3: **5way-1shot and 5way-5shot classification accuracy in augmented few-shot classification task** and 95% confidence interval on Mini-ImageNet and Tiered-ImageNet (over 2000 tasks), using Conv4 as the backbone.the terms "S" and "W" denote the strong-level and weak-level augmentation strategies applied to the support data in the meta-test phase.

| | Mini-ImageNet (S) | | Mini-ImageNet (W) | |
|---|---|---|---|---|
| Methods | 1-Shot | 5-Shot | 1-Shot | 5-Shot |
| MAMLFinn et al. (2017) | 28.13 | 37.77 | 35.89 | 49.54 |
| LLC + MAML | **30.64** | **40.79** | **37.11** | **50.38** |
| Unicorn-MAMLYe & Chao (2021) | 29.26 | 40.58 | 36.07 | 51.43 |
| LLC + Unicorn-MAML | **31.37** | **42.59** | **38.94** | **54.11** |

Table 4: **5way-1shot and 5way-5shot classification accuracy in strongly augmented few-shot classification task** and 95% confidence interval on Mini-ImageNet and Tiered-ImageNet (over 2000 tasks), using ResNet-12 as the backbone.

| | Mini-ImageNet(S) | | Tiered-ImageNet(S) | |
|---|---|---|---|---|
| Methods | 1-Shot | 5-Shot | 1-Shot | 5-Shot |
| MAMLFinn et al. (2017) | 49.94 | 73.46 | 51.87 | 75.11 |
| LLC + MAML | **57.31** | **78.32** | **55.79** | **77.29** |
| Unicorn-MAMLYe & Chao (2021) | 50.57 | 73.68 | 53.01 | 76.08 |
| LLC + Unicorn-MAML | **57.75** | **77.25** | **56.39** | **79.16** |

outcomes across different domains. We achieved optimal performance on five datasets and second-best performance on three additional datasets. Notably, our approach demonstrated a strong lead on the Cars, EuroSAT, ISIC, and CropDisease datasets. This suggests that LLC also demonstrates strong generalization in cross-domain few-shot problems, reducing the impact of shortcut features during the fast adaptation phase.

### 5.2.3 AUGMENTED FEW-SHOT LEARNING PROBLEMS.

**Augmented few-shot accuracy.** Table. 3 presents the performance of Conv4 on the Mini-ImageNet dataset under varying levels of augmentation. LLC has an approximate 2% increase in classification accuracy on query data, irrespective of whether the perturbations are weak or strong. Table.4 demonstrates the performance of ResNet-12 under strong augmentation on both Mini-ImageNet and Tiered ImageNet datasets, where LLC has a approximate average 4.5% in classification accuracy. It is evident that LLC confers greater improvements on models with larger capacities and contributes to a significant increase in accuracy for various tasks.

.

Table 5: **Learning consistency in strong augmented few-shot classification task** on Mini-ImageNet and Tiered-ImageNet.

| Methods | Mini-ImageNet | | Tiered-ImageNet | |
|---|---|---|---|---|
| | 1-Shot | 5-Shot | 1-Shot | 5-Shot |
| MAML Finn et al. (2017) | 85.88 | 94.03 | 84.93 | 93.87 |
| LLC + MAML | **98.58** | **99.00** | **99.70** | **99.80** |
| Unicorn-MAML Ye & Chao (2021) | 87.55 | 94.60 | 86.67 | 95.41 |
| LLC + Unicorn-MAML | **99.91** | **99.92** | **99.94** | **99.96** |

Table 6: **Ablation study on Mini-ImageNet.** All models are trained on the meta-train set of Mini-ImageNet (over 2000 tasks).

| $\mathcal{A}ug$ | $\mathcal{L}_{CK}$ | Mini-ImageNet | |
|---|---|---|---|
| | | 1-shot | 5-shot |
| ✗ | ✗ | $64.43 \pm 0.46$ | $83.90 \pm 0.29$ |
| ✓ | ✗ | $64.31 \pm 0.48$ | $84.14 \pm 0.28$ |
| ✓ | ✓ | $\mathbf{65.41} \pm \mathbf{0.47}$ | $\mathbf{84.88} \pm \mathbf{0.29}$ |

Table 7: The result of accuracy and learning consistency for different $\gamma$.

| $\gamma$ | 0.1 | 0.3 | 0.5 | 0.8 | 1.0 | 2.0 | 3.0 |
|---|---|---|---|---|---|---|---|
| Accuracy | 83.71 | 84.05 | 84.31 | 84.55 | **84.88** | 84.86 | 84.76 |
| Consistency | 95.13 | 97.66 | 98.25 | 98.87 | 99.00 | 99.09 | 99.10 |

Table 8: **Ablation study on different augmentation strategies.** RR denotes Random Crop, CJ denotes Color Jitter, GC denotes Grayscale Conversion, GB denotes Gaussian Blur, and RHF denotes Random Horizontal Flip. All models are trained on the meta-train set of Mini-ImageNet.

| RR | CJ | GC | GB | RHF | Mini-ImageNet | |
|---|---|---|---|---|---|---|
| | | | | | 1-shot | 5-shot |
| ✗ | ✓ | ✓ | ✓ | ✓ | 65.13 | 84.50 |
| ✓ | ✗ | ✓ | ✓ | ✓ | 65.21 | 84.63 |
| ✓ | ✓ | ✗ | ✓ | ✓ | 65.18 | 84.67 |
| ✓ | ✓ | ✓ | ✗ | ✓ | 65.02 | 84.53 |
| ✓ | ✓ | ✓ | ✓ | ✗ | 65.32 | 84.71 |
| ✓ | ✓ | ✓ | ✓ | ✓ | **65.41** | **84.88** |

**Learning consistency.** Table.5 presents the learning consistency of different initialized models, as quantified by Eq.6. It is observed that both MAML and its variant, MAML-Unicorn, tend to learn inconsistent knowledge in both 5way-1shot and 5way-5shot scenarios. This implies that the model initialized by MAML and Unicorn-MAML is easily influenced by the different shortcut features produced by different augmentations, while our method achieves around 99% consistency in knowledge across both datasets for 5way-1shot and 5way-5shot problems. The result shows that our method significantly enhances the model's ability to learn consistently.

5.3 ABLATION STUDY

To further explore the effectiveness of LLC, we conducted some ablation studies on LLC. We focus on the affection of data augmentation, the number of inner steps, and the similarity function.

**Impact of $\gamma$.** We further investigate the effect of different values of $\gamma$, with results presented in Table.7. As $\gamma$ increases, we observe a consistent improvement in both learning consistency and accuracy. However, excessively large values of $\gamma$ can hinder model convergence, leading to a slight degradation in performance. Based on these observations, we set $\gamma = 1$ as the optimal choice.

**The impact of data augmentation and $\mathcal{L}_{CK}$.** Table.6 illustrates the impact of data augmentation and $\mathcal{L}_{CK}$. The first row presents the results of LLC without data augmentation and $\mathcal{L}_{CK}$, which is equivalent to MAML. The second row shows the results of LLC without $\mathcal{L}_{CK}$, which is equivalent to MAML with augmentation. The third row displays the results of LLC. The result indicate that augmentation is not the primary factor in LLC's improvement. The main improvement is attributed to $\mathcal{L}_{CK}$, which enables the initialized model to learn consistently. This result further underscores the motivation to learn consistently. We further invastigate how different augmentation strategies influence the performance in Table.8. The results indicate that LLC exhibits robustness across different augmentation methods.

**The impact of the inner step.** We further investigated the impact of different inner steps during the meta-test phase on the model's few-shot classification accuracy and precise learning capabilities. Fig.4 illustrates the impact of the number of inner steps during the meta-test phase on the performance of the LLC algorithm. The results indicate that for any given number of inner steps,

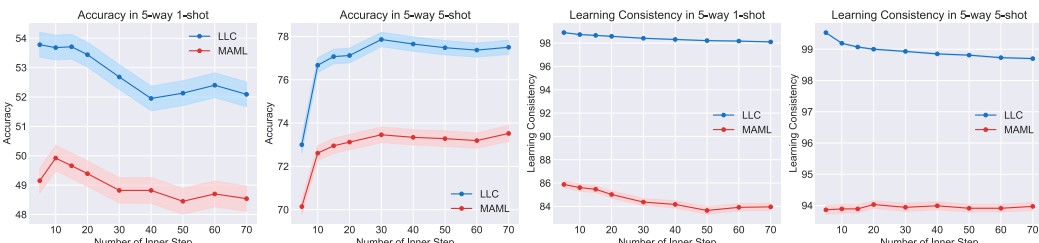

Figure 4: **The 5way-1shot and 5way-5shot classification accuracy and the consistency of learned knowledge with different numbers of inner steps** with 95% confidence interval, averaged over 2000 tasks

Table 9: **The result with different** $\mathcal{F}_{sim}$ **in standard few-shot classification task on Mini-ImageNet,** using ResNet-12 as the backbone.

| $L_{ck}$ | Cosine | KL | JS | NMSE |
|---|---|---|---|---|
| 1-shot | **65.41** | 65.10 | 64.77 | 65.02 |
| 5-shot | **84.88** | 84.73 | 84.29 | 84.49 |

the models trained using LLC consistently outperformed those trained with MAML. Specifically, in the 5way-1shot and 5way-5shot tasks, LLC achieved an accuracy of approximately 7% and 4% higher than MAML, respectively. Concerning the consistency of the knowledge learned, there was a trend of decreasing consistency for both MAML and LLC as the number of inner steps increased. This suggests that an excessive number of inner steps during the meta-test phase may lead to the model learning shortcut features. However, LLC still maintained approximately 99% consistency in different settings of the inner step, which shows the robustness and generalization of LLC.

**The impact of** $\mathcal{F}_{sim}$**.** Table.9 demonstrates the effect of different similarity functions on the performance of LLC. All the similarity function yield strong results, while cosine reach the highest result. This suggests that overly strict constraints, as imposed by NMSE, may adversely affect classification performance. Therefore, cosine similarity is recommended for use in LLC.

## 5.4 FURTHER ANALYSIS

**Compute consumption.** Compared to MAML, LLC achieves parity in algorithmic complexity by substituting different tasks with varied versions of the same task. Consequently, the computational overhead of LLC aligns with that of MAML.

**Visualization.** To gain deeper insights into the impact of LLC on the learning capabilities of models, we visualized the models updated with augmented data, as shown in Fig.5 in the Appendix. Specifically, during the meta-test phase, we visualized models trained with MAML and LLC. The model was first fine-tuned with support data, with the number of inner steps set to 20. Then, query data was employed as the visualized data. Grad-CAM++(Chattopadhay et al., 2018) was utilized to visualize the critical regions that the models focused on for understanding the query data. The visualizations reveal that the model trained with MAML is more likely influenced by the environment, whereas model trained with LLC concentrate more on the objects pertinent to classification.

## 6 CONCLUSION

A key challenge in few-shot learning lies in the tendency of models to rely on shortcut features. In this work, we observe that models trained with higher learning consistency are less susceptible to such shortcuts. Motivated by this observation, we propose a meta-learning framework termed Learn to Learn Consistently (LLC). LLC updates the model in the inner loop using different augmented views of the support set, and subsequently maximizes the consistency of the outputs for the same query across these differently updated models. We evaluate LLC on three few-shot learning benchmarks and show that it consistently delivers substantial performance gains across diverse scenarios. By encouraging consistency in the learning process, LLC offers a novel perspective in meta-learning and represents a step forward in enhancing the generalization ability of few-shot learners.

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

## A    MORE EXPERIMENTAL RESULTS

### A.1    RESULT OF APPLYING $L_{ck}$ TO LATENT

Table.10 reports the results when the consistency loss $L_{ck}$ is applied to the latent representations. The result shows that applying to logits is better than to latent representation. Since our objective is to learn consistent knowledge—defined in terms of the distribution of the logits—it is preferable to apply $L_{ck}$ directly to the logits.

Table 10: **Apply $L_{ck}$ to latents**

| $L_{ck}$ | Logits | Latents |
|---|---|---|
| Accuracy (%) | **84.88** | 84.29 |

### A.2    RESULTS OF VIT

We compare our approach with those proposed in Hiller et al. (2022); Hao et al. (2023). Although CPEA Hao et al. (2023) is specifically designed for ViT architectures, our method achieves comparable performance while outperforming FewTURE Hiller et al. (2022). These results further validate the effectiveness and generalizability of our approach.

Table 11: **More results compared with SOTA**

| | | Mini-ImageNet | | Tiered-ImageNet | |
|---|---|---|---|---|---|
| Methods | Model | 1-Shot | 5-Shot | 1-Shot | 5-Shot |
| FewTUREHiller et al. (2022) | ViT-S/16 | 68.02 | 84.51 | 72.96 | 86.43 |
| CPEAHao et al. (2023) | ViT-S/16 | **71.97** | 87.06 | **76.93** | 90.12 |
| LLC (Ours) | Res-12 | 65.41 | 84.88 | 68.51 | 86.87 |
| LLC (Ours) | ViT-S/16 | 70.69 | **88.43** | 75.86 | **90.97** |

## B    HYPERPARAMETERS AND CODE ENVIRONMENT OF THE EXPERIMENT

**Hyperparameters.** The hyperparameters has shown in the Table.12, Table.13, Table.14.
**Calculation resources and Environment.** Our experiment is conducted on NVIDIA A800 80GB PCIe and NVIDIA A100 40GB PCIe. We use Python version 3.10.14, PyTorch version 2.3.0, and CUDA toolkit 12.1 on A800 80GB, and use Python version 3.11.9, PyTorch version 2.3.0, and CUDA toolkit 11.8 on A100 40GB.

Table 12: Experimental Setup

| Parameter | Value |
|---|---|
| task batch Size | 4 |
| inner loop learning rate | 0.05 |
| outer loop learning rate | 0.001 |
| outer data points | 15 |
| outer loop learning rate decay | 1/10 every 10 epochs |
| coefficient $\gamma$ | 1 |

## C    ALGORITHM

The specific algorithm flow of LLC is shown in Algo.2

Table 13: Augmentations for Strong-Augmented Few-Shot Scenario

| Augmentation | Parameters | Probability |
|---|---|---|
| Random Resize | scale: 0.5–1 | - |
| Color Jitter | (0.8, 0.8, 0.8, 0.2) | 0.8 |
| Grayscale Conversion | - | 0.2 |
| Gaussian Blur | $\mathbb{E}$: 0.1, $\sigma^2$: 2 | 0.5 |
| Random Horizontal Flip | - | 0.5 |

Table 14: Augmentations for Weak-Augmented Few-Shot Scenario

| Augmentation | Parameters | Probability |
|---|---|---|
| Center Crop | $84 \times 84$ | - |
| Color Jitter | (0.4, 0.4, 0.4, 0.1) | 0.8 |
| Grayscale Conversion | - | 0.2 |
| Gaussian Blur | $\mathbb{E}$: 0, $\sigma^2$: 1 | 0.5 |
| Random Horizontal Flip | - | 0.5 |

## D  COMPUTE COMSUMPTION

We counted the training time of LLC and MAML during the meta-train phase. Specifically, one epoch includes the optimization of 100 batches, where MAML uses 4 tasks for each batch for optimization, while LLC uses 1 task and enhances each batch 4 times for optimization. LLC has the same complexity as maml and thus has similar optimization times.

Table 15: **The training time of LLC and MAML during the meta-train phase**

| Time(Min) | Mini-ImageNet | Tiered-ImageNet |
|---|---|---|
| MAML | 2.61 | 2.67 |
| LLC (Ours) | 2.76 | 2.85 |

## E  FURTHER VISUALIZATION

In Figures 5, we present visualizations of our method using Grad-CAM++. These visualizations indicate that the model trained with MAML exhibits a greater susceptibility to environmental influences. In contrast, the model trained with LLC demonstrates a focused attention on classification-relevant objects and exhibits an enhanced ability to recognize a broader range of features.

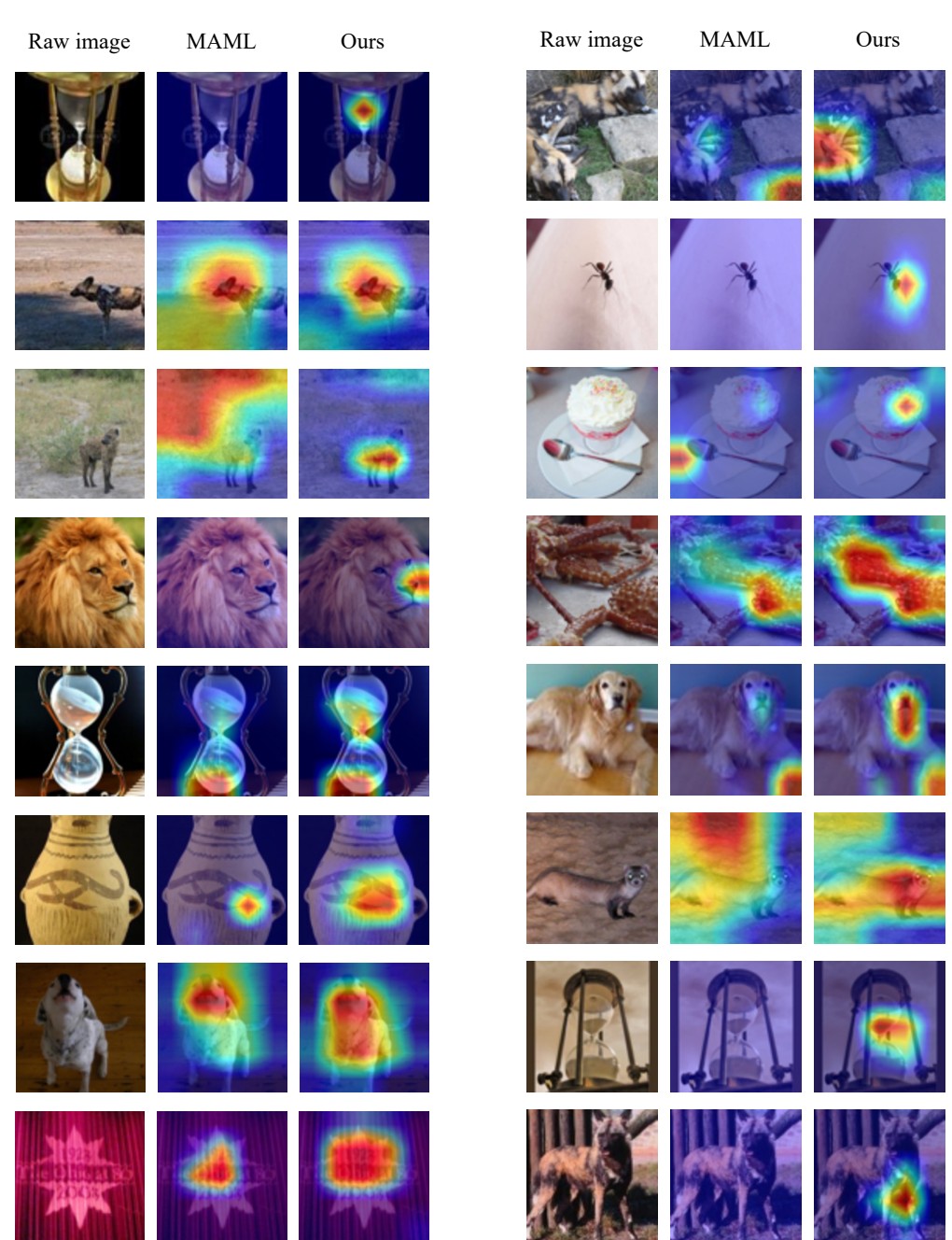

Figure 5: **The results of the visual analysis** on the test set of *Mini*ImageNet with MAML and LLC. The left shows the main results, while the right presents additional results.

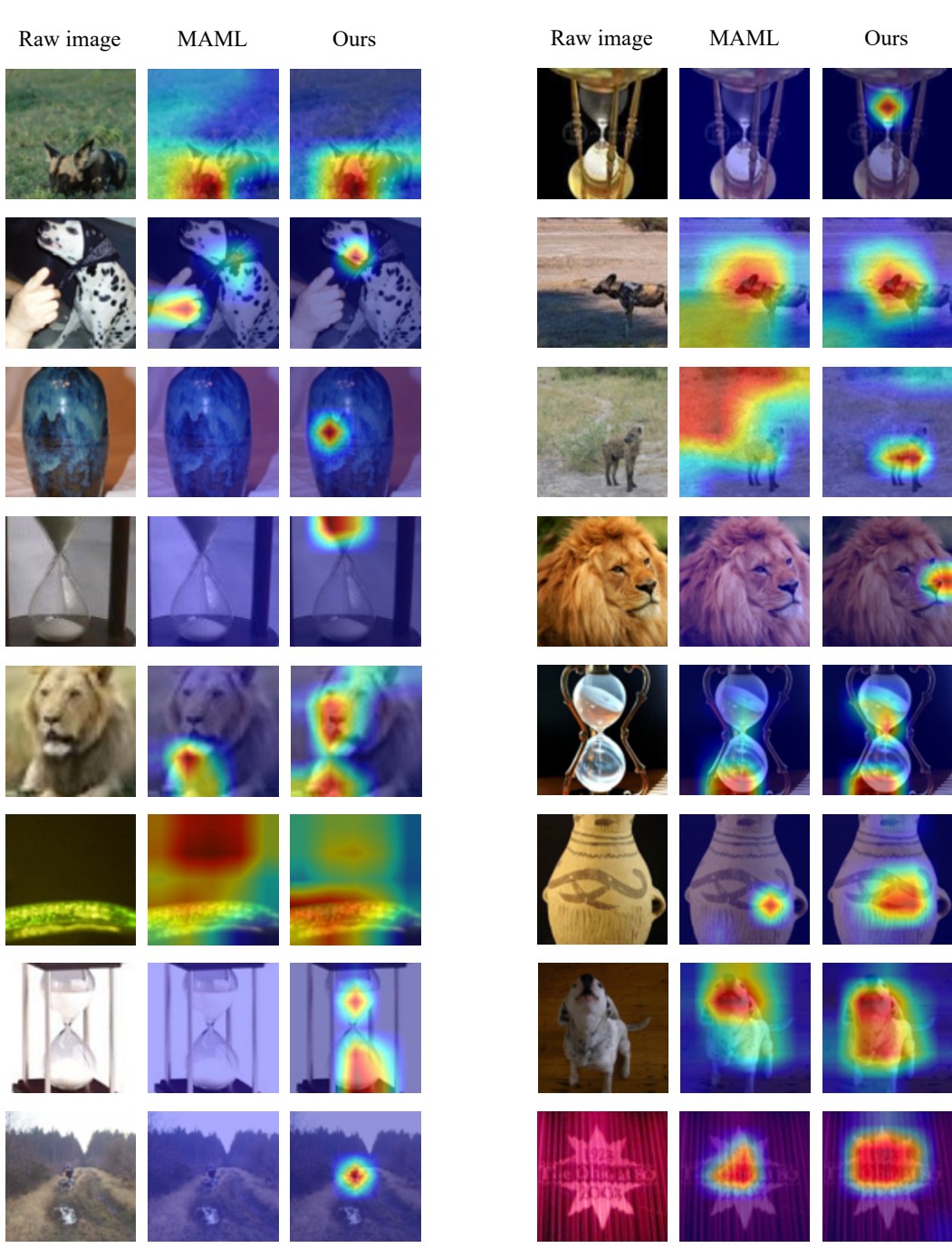

Figure 6: **The results of the visual analysis** on the test set of *Mini*ImageNet with MAML and LLC. The left shows the main results, while the right presents additional results.

---

**Algorithm 1:** Evaluate the learning consistency and accuracy of model initialized by MAML

---

**Given** the learned initialization $\theta$ by MAML
**for** $t \in \{1, \cdots, T\}$ **do**
    **Sample** task $\mathcal{T} = (\mathcal{S}, \mathcal{Q}) \sim \mathcal{D}_{\text{meta-test}}$
    **for** $i \in \{1, \cdots, n\}$ **do**
        **Random Augmented** the support data: $Get \; \mathcal{A}ug^i(\mathcal{S})$
        **Update** $\theta$ by augmented support data $\mathcal{S}_i$: $\theta_i = \mathcal{U}^k(\theta, \mathcal{A}ug^i(\mathcal{S}))$
        **Get the output** of the query data $x_q$ by $\theta_i$: $v_i = f_{\theta_i}(x_{\mathcal{Q}})$
    **end**
    **Record** the consistency and average accuracy of the
    output $\{v_i\}$:
    $\mathcal{C}[t] = \frac{1}{n} \sum_{i=1}^{n} \mathcal{F}_{\text{sim}} \left( v_i, \frac{1}{n} \sum_{i=j}^{n} v_j \right)$
    $\mathcal{A}[t] = \frac{1}{n} \sum_{i=1}^{n} \mathcal{A}cc(v_i)$
**end**
**Return** $\mathcal{C}, \mathcal{A}$

---

**Algorithm 2:** Learn to Learn Consistently

---

**Given** the learned initialization $\theta^0$ pretrained on meta-train set
**for** $t \in \{1, \cdots, T\}$ **do**
    **Sample** task $\mathcal{T} = (\mathcal{S}, \mathcal{Q}) \sim \mathcal{D}_{\text{meta-train}}$
    **for** $i \in \{1, \cdots, n\}$ **do**
        **Random Augmented** the support data: $Get \; \mathcal{A}ug^i(\mathcal{S})$
        **Update** $\theta^{t-1}$ by augmented support data $\mathcal{S}_i$: Get $\theta_i^{t-1} = \mathcal{U}^k(\theta^{t-1}, \mathcal{A}ug^i(\mathcal{S}))$
        **Get the output** of the query data $x_q$ by $\theta_i^{t-1}$: Get $v_i = f_{\theta_i^{t-1}}(x_{\mathcal{Q}})$
    **end**
    **Calculate** the outer loop loss: $\mathcal{L}_{\text{total}} = \mathcal{L}_{\text{cls}} + \gamma \cdot \mathcal{L}_{\text{CK}}$
    **Update** the parameters by outer loop loss: $\theta^t = \theta^{t-1} - \beta \cdot \nabla_{\theta^{t-1}} \mathcal{L}_{\text{total}}$
**end**
**Return** $\theta^T$

---