# OpenReview forum: "Learn to Learn Consistently via Meta Self-distillation for Few-shot Classification"
_ICLR.cc/2026/Conference — ICLR 2026 Conference Withdrawn Submission_

### Official Review · Reviewer_ZRio · 2025-10-28

**Soundness:** 3
**Presentation:** 3
**Contribution:** 2
**Rating:** 2
**Confidence:** 5

**Summary:**

This paper aims to solve the few-shot classification problem by employing self-distillation method, where the authors facilitate generalization through enforcing learning consistency. They prevent learning shortcut features for novel tasks by finding a more generalizable initialization, where they employ data augmentation to generate different views of the same data and maximize the learning consistency to find a better encoder. Experiments on Mini-ImageNet, Tiered-ImageNet, and several cross-domain benchmarks show consistent improvements over MAML and its variants.

**Strengths:**

- The formulation for the proposed method is simple and reasonable, and contributions claimed by the authors are straightforward and clear.
- The implementation is simple and can be easily reproduced by extending upon numerous MAML variants.
- The results show consistent improvements on numerous few-shot classification datasets and various MAML varints, with ablation experiments to show some empirical performance gains.

**Weaknesses:**

- The concept of enforcing consistency and employing multi-view learning through data augmentation has been introduced in numerous previous works [a,b,c,d], and these works should be cited and throughly compared.

  - [a] Prompt Learning via Meta-Regularization, Park et al., CVPR 2024 : Gradient-level alignment to enforce consistency between different tasks
  - [b] Consistent Meta-Regularization for Better Meta-Knowledge in Few-Shot Learning, Tian et al., TNNLS 2022 : Consistency is enforced in terms of training and test data distribution discrepency
  - [c] DAC-MR: Data Augmentation Consistency Based Meta-Regularization for Meta-Learning, Shu et al., arXiv:2305.07892 : Also regularizes model by enforcing data augmentation consistency
  - [d] MetaMix: Improved Meta Learning with Interpolation based Consistency Regularization, Chen et al., ICPR 2020 : Uses Mixup technique for data augmentation, and model regularization

- Based on the previously proposed methods based on semi-supervised learning literature, this paper's contributions seem to simply apply the techniques used in the previous methods, without adding insightful modifications and analysis for meta-learning and few-shot learning tasks.

- Simply employing cosine similarity and enforcing the metric on the output logits seem to be a superficial way to enforce consistency, and further experiments on enforcing consistency by feature map distillation, gradient direction matching, etc. would be a straightforward extension.

- Although the improvements in empirical performance were consistent, they are very marginal and it limits the grounds for the contribution claimed by the authors.

- Although in the introduction section (P2L77) the authors state that their method was tested under regression settings, there are no experiments for the regression tasks in the experiment section.

**Questions:**

Please refer to the weaknesses section. Although the proposed scheme is simple and straightforward, limitations in the contributions and minor performance gains should be addressed.

---

### Official Review · Reviewer_HesL · 2025-10-28

**Soundness:** 1
**Presentation:** 1
**Contribution:** 2
**Rating:** 2
**Confidence:** 4

**Summary:**

This paper recognizes an issue for meta-learning algorithms: the exploitation of "short-cut" features for few-shot task adaptation, without acquiring generalization ability across unseen tasks. Then, the authors propose to enforce the notion of learning consistency throughout the meta-training process.

Unfortunately, the motivation and the notion of "short-cut" features, the learning consistency, and its distinction from the existing contrastive learning or self-distillation are not sound or rigorous. The presentation quality throughout the paper also need more attention and effort to reach the academic level.

**Strengths:**

This paper examines the tendency of meta-learners to exploit "short-cut" features for quick solving few-shot tasks during the meta-training stage, and are therefore unable to generalize to meta-testing tasks when the class concepts changed. The authors drew inspiration from contrastive learning and self-distillation to propose the notion of learning consistency, which is explicitly enforced within the meta-learning process for prompting better generalization ability.

**Weaknesses:**

As part of the fundamental motivation of the paper, the "short-cut" features are not sound or rigorously defined. Specifically, the authors failed to clarify what makes a specific feature as "short-cut", and what features should be considered generalized? The example short-cut features (i.e. background, noise shape, color) could reasonably be considered generalized depending on the task settings. As the authors dive into the proposed learning consistency, the short-cut features are implicitly assumed to be the features that would be changed from data augmentation operations, which is not justified.

The proposed notion of learning consistency is also questionable: while the authors attempted to make the distinction that the proposed learning consistency is different from prior concepts such as the learning objective from contrastive learning, the learning consistency metric $LC(\theta)$ is yet still defined on the model outputs given augmented inputs, instead of truly on the model update path or parameters, making it very similar to contrastive learning in root.

The presentation and type-setting could be largely improved throughout the paper, with examples including but not limited to the following:
1. Under Section 2.2: "Different from self-distillation that directly aligns representations, our approach focuses on aligning the models differently updated by different augmented Samples." The word "Samples" shouldn't be capitalized.
2. Within Equation (1), the general definition of the loss function $L_{FSL}$ is not provided.
3. Within Definition 1, the sentence "Consider two random data augmentation operators...", at the end it should be $Aug_j(S)$, with the suffix "j" and the period mark missing at the end. Right at the next sentence, "These augmented data are then used to..." should be "is then used to" as data is not countable.
4. In the last sentence prior to Section 4.2: "[...] the parameters will learn consistent knowledge from
different augmented views of the same image, thus less influenced by the shortcut features." There is a missing be verb before "less influenced by...".
5. In Equation (6), "n" is used for both inner and outer sum.
Though each of these is minor, as they accumulate throughout the paper, the presentation quality is largely degraded.

**Questions:**

For the confidence intervals presented within the experiment section, are they based on experiments on multiple random seeds, or are just directly computed as the standard deviation over the task accuracies over all tasks under one seed?

What are some speculations the authors could pose on the setups where LLC falls short against some baselines? What are the unique natures of these trails compared against the majority trails leading to the LLC not performing the best?

For experiments under augmented few-shot learning, are the augmentations only applied to the query samples in meta-training tasks, or they are also applied to the meta-testing tasks for accuracy evaluations?

---

### Official Review · Reviewer_z6rk · 2025-10-29

**Soundness:** 3
**Presentation:** 2
**Contribution:** 1
**Rating:** 2
**Confidence:** 5

**Summary:**

In few-shot learning, a key challenge is shortcut bias, where models overfit to spurious cues from the limited support set and fail to generalize to novel tasks. To address this, this paper proposes Learn to Learn Consistently (LLC), a simple yet effective meta-learning framework that enforces learning consistency during training.

**Strengths:**

This paper proposes Learn to Learn Consistently (LLC), a simple yet effective meta-learning framework that enforces learning consistency during training. Extensive experiments validate the effectiveness of our approach.  Besides, this paper provided enough visulations to show the contributions.

**Weaknesses:**

1.	Few-shot learning has been a well-explored research topic for decades, with substantial progress achieved. As a result, the contribution of this paper seems somewhat limited in scope.
2.  I carefully observed the experimental results in Tables II and III and found that the improvements are marginal. As mentioned earlier, the overall contribution of this paper appears limited.
3. In Table VIII, I did not observe noticeable differences among the various augmentation strategies. A more detailed analysis would be helpful.
4. Since MAML training is known to be unstable, I would like to see more details regarding the training process and convergence behavior.
5. Moreover, LLC does not achieve the best performance in the 1-shot setting on the Tiered-ImageNet dataset as shown in Table I. This appears to be an error.

**Questions:**

please see the weakness. I would like to see the more details and novelty.

---

### Official Review · Reviewer_R5xg · 2025-11-04

**Soundness:** 3
**Presentation:** 2
**Contribution:** 3
**Rating:** 4
**Confidence:** 3

**Summary:**

This paper introduces Learn to Learn Consistently (LLC), an optimization-based meta-learning framework for few-shot classification. The method claims that the learning consistency: the degree to which a model produces similar predictions when trained on different augmentations of the same support set.

It was built on the top of MAML:
In the inner loop, multiple augmented versions of each task are used to independently update model parameters.
In the outer loop, consistency across these updated models is enforced via a cosine-similarity loss on the shared query set, combined with a standard classification loss.

Extensive experiments on multiple datasets on classic few shot learning and several cross-domain benchmarks demonstrate that LLC achieves consistent improvements over baselines.

**Strengths:**

1. The motivation is clear.
2. The proposed idea of learning consistency is pretty simple.
3. Extensive experiments on classic few shot learning and several cross-domain benchmarks demonstrate that LLC achieves consistent improvements over baselines.

**Weaknesses:**

1. The theoretical insight is limited: While the definition of learning consistency is formal, the property 1 appears to be more of an extended definition than a rigorously established property. Providing an analytical bound or theoretical justification would significantly strengthen the paper’s claims. Also, its connection to generalization and shortcut mitigation is not analytically justified.

2. The complexity analysis is limited: Although the paper claims that "*Compared to MAML, LLC achieves parity in algorithmic complexity by substituting different tasks with varied versions of the same task.*", however, this statement is not supported by any quantitative or experimental evidence.

3. From the perspective of experiments, the paper could emphasize the key components: self-distillation and data augmentation, more directly.
- 3.1 For "self-distillation", it would be informative to compare LLC against established self-distillation approaches (e.g., BYOL, SimSiam) to better contextualize its novelty or effectiveness.
- 3.2 For data-augmentation, there are many different approaches, why does the paper select these five (RR, CJ, GC, GB, RHF)? The rationale for choosing the five augmentations is not discussed. Exploring alternative or domain-specific augmentations, or analyzing the sensitivity of LLC to these choices would provide deeper insight into the robustness and generality of the approach.

4. Presentation issue: While the paper is understandable for readers familiar with meta-learning, the notation can be difficult to follow for others. Also, the inconsistency of final equations (L_cls and L_ck) in Figure 3 and Equ. 7 will be an important weakness. Others like minor typos in Definition 1 for Aug_j (S), etc.

**Questions:**

See above weakness.

---

### Note · Authors · 2025-11-13

I have read and agree with the venue's withdrawal policy on behalf of myself and my co-authors.